# Continual Learning with Orthogonal Weights and Knowledge Transfer

## Abstract

Orthogonal projection has been shown highly effective at overcoming *catastrophic forgetting* (CF) in continual learning (CL). Existing orthogonal projection methods are *all* based on *orthogonal gradients* (OG) between tasks. However, this paper shows theoretically that OG cannot guarantee CF elimination, which is a major limitation of the existing OG-based CL methods. Our theory further shows that only the *weight/parameter-level orthogonality* between tasks can guarantee CF elimination as the final classification is computed based on the network weights/parameters only. Existing OG-based methods also have two other *inherent limitations*, i.e., *over-consumption of network capacity* and *limiting knowledge transfer* (KT) across tasks. KT is also a core objective of CL. This paper then proposes a novel *weight-level orthogonal projection* method (called STIL), which ensures that each task occupies a weight subspace that is orthogonal to those of the other tasks. The method also addresses the two other limitations of the OG-based methods. Extensive evaluations show that the proposed STIL[1] not only overcomes CF better than baselines, but also, perhaps more importantly, performs KT much better than them.

## 1 Introduction

Continual learning (CL) using deep neural networks (DNNs) to learn a sequence of tasks is a challenging problem. Two key issues are overcoming *Catastrophic Forgetting* (CF) (McCloskey & Cohen, 1989; Ratcliff, 1990), and transferring the previously learned knowledge to the new task, namely *Knowledge Transfer* (KT). Most existing CL methods focuses on mitigating CF, which can be generally divided into network expansion methods and non-expansion methods. For example, LwF Li & Hoiem (2018), CGN (Rusu et al., 2016), DEN (Yoon et al., 2018), APD (Yoon et al., 2020), and BNS (Qin et al., 2021) are representative expansion methods. They expand the network for each task to overcome CF, but they suffer from memory explosion with more tasks learned. While the basic idea of non-expansion methods is to constrain the gradient update of the network weights towards less harmful directions to protect the previously learned knowledge. Among the non-expansion methods, orthogonal projection methods that ensure weights modification of **orthogonal gradients** between tasks have been shown effective in overcoming CF. Our work is closely related to these methods.

However, orthogonal gradients (**OG**) methods have three major limitations. **(1) Unable to eliminate CF.** This paper theoretically shows (for the first time) that ***orthogonal gradients cannot guarantee CF elimination***. Only ***orthogonal-weight**/parameter* (**OW**) between tasks can eliminate CF as the final classification is computed based on the network weights or parameters only. Clearly, gradient-level orthogonality between tasks does not guarantee weight-level orthogonality (see Theorem 1 and Sec.5). Note that we will use *weight* and *parameter* interchangeably from here onward. **(2) Over-consumption of network capacity.** Since each task is learned in an OG space, only a small number of tasks can be learned in a network (see Theorem 2). **(3) Limited KT**. As each task is in a separate orthogonal space, there is little sharing of knowledge across tasks, leading to little KT.

This paper proposes a novel *Self-adaptive Task Incremental Learning* (**STIL** in short) to solve these three problems. Task-incremental learning (TIL) is one of the important settings of CL (van de Ven & Tolias, 2019). The other one is *class-incremental learning* (CIL). The two settings, i.e., TIL and CIL, are suitable for different types of applications.

---

[1]The code of the proposed method STIL can be found at https://github.com/STILalg/STIL

To solve problem **(1)** above, we introduce the concept of **Orthogonal Important Weight Subspace** (**OIWS**). It enables each new task to occupy its own important weight subspace, and all the weight subspaces between tasks are orthogonal to each other. The paper then proposes a novel OIWS-based TIL strategy, which ensures that the weight updates for each task are performed in its own OIWS so as to ensure the orthogonal weights between tasks maximally eliminate CF. Instead of using the full weight sub-space for each task, STIL uses the subspace of important weights (see Sec. 4.1 for details), which allows STIL to learn a large number of tasks, which solves problem **(2)** above.

In the TIL setting, transferring knowledge across tasks is a major goal (Ke et al., 2020). When learning a new task $t$, naturally some previously learned tasks may be similar to $t$, then the knowledge from them should be leveraged to learn $t$ better (i.e., *forward knowledge transfer*, FKT). Conversely, the learning of $t$ may also improve those similar previous tasks (i.e., *backward knowledge transfer*, BKT). This work wants to achieve FKT and BKT across similar tasks to address the problem **(3)**.

Although some existing TIL methods perform KT, e.g., CAT (Ke et al., 2020) using an additional sub-model, and TRGP (Lin et al., 2022a) and CUBER (Lin et al., 2022b) using layer-wise scaling matrices, they still have some major shortcomings, e.g., only having limited FKT or no BKT, e.g., TRGP. To maximumly enable KT, this paper proposes a new task similarity detection method based on only direct reuse of the knowledge already learned in the current network and a new cross-entropy loss with constraints. Equipped with the proposed OIWS and the above two techniques, the proposed STIL can effectively overcome CF and perform KT.

Extensive experiments show that the proposed STIL method not only overcomes CF better than existing state-of-the-art baselines on dissimilar, similar and mixed tasks sequences, but also, perhaps more importantly, performs KT dramatically better than them when similar tasks are learned.

## 2 RELATED WORK

This paper focuses on task-incremental learning (TIL) without network expansion. Existing non-expansion TIL methods can be divided into the following categories: *Regularization based* methods, e.g., EWC (Kirkpatrick et al., 2017) and UCL (Ahn et al., 2019), penalize modifications to important weights of old tasks through regularizations. *Experience-replay based* methods, e.g., iCaRL (Rebuffi et al., 2017) and GEM (Lopez-Paz & Ranzato, 2017), overcome CF by replaying the data (either a sample of the real data or generated data) of old tasks in learning the new task. *Orthogonal-projection based* methods, e.g., OWM (Zeng et al., 2019), OGD (Farajtabar et al., 2020), GPM (Saha et al., 2021), TRGP (Lin et al., 2022a), and CUBER (Lin et al., 2022b), update the weights with gradients in the orthogonal directions of old tasks, without using any old task data. *Parameter isolation methods* like HAT (Serrà et al., 2018) and SupSup (Wortsman et al., 2020) isolate a sub-network for each task. There are also reinforcement learning based (Kaplanis et al., 2019; Qin et al., 2021), soft mask based (Konishi et al., 2023) and meta-learning based methods (Rajasegaran et al., 2020).

**Orthogonal Projection Methods.** These methods are closely related to our work. The approach was first proposed in OWM for overcoming CF, which ensures *orthogonal gradients* (OG) in its weight modification in the span space of the input data. After OWM, ORTHOG-SUBSPACE (Chaudhry et al., 2020), OGD, GPM, TRGP and CUBER were proposed. OGD restricts the direction of gradient updates to be orthogonal to the gradients of old tasks. GPM makes OWM more efficient, which is inherited by TRGP and CUBER. They all learn each task by taking the ***OG*** direction to the gradient subspaces for the old tasks. These OG-based methods have been shown effective in mitigating CF but not eliminating CF as we will see in Theorem 1 (Sec. 3.1). Since we have discussed this and two other weaknesses of these methods in Sec. 1, we don't repeat them here.

**Knowledge Transfer (KT).** Several early non-neural network based methods have done KT among similar tasks using KNN (Thrun, 1998), regression (Ruvolo & Eaton, 2013), and naive Bayes (Mitchell et al., 2018; Riemer et al., 2019), but they do not deal with CF. A few DNN based methods like CAT, TRGP and CUBER simultaneously deal with both CF and KT. CAT uses binary masks of neurons in HAT to achieve CF prevention, and employs a separate model to perform task similarity detection for KT. TRGP first selects the most related old tasks within the 'trust region' for the new task, and then reuses the frozen weights in layer-wise scaling matrices to jointly optimize the matrices and the model to achieve FKT. On the basis of TRGP, CUBER first analyzes the conditions under which updating the learned model of old tasks could be beneficial for CL and lead to BKT. It then

proposes a new method with FKT and BKT. The main weaknesses of CAT, TRGP and CUBER are their limited FKT or negative BKT leading to CF again in their BKT (see the experiments in Sec.5).

Contrary to the existing KT-based methods, STIL stores only the most basic information to efficiently deal with both CF and KT. Based only on reusing the knowledge/weights of the learned tasks in the current model and the Wasserstein' distances (Panaretos & Zemel, 2019) of the model losses between a new task and old tasks, STIL easily detects task similarity. The proposed cross-entropy loss with constraints and its solution enables STIL to efficiently transfer knowledge forward and backward.

## 3 THEORETICAL ANALYSIS OF THE CF AND KT PROBLEMS

**Task Incremental Learning (TIL).** Let $T$ tasks be $\mathbb{T} = \{t\}_{t=1}^{T}$, which are learned sequentially. Each task has a training dataset with its task descriptor $t$, $\mathbb{D}_t = \{((\boldsymbol{x}_{t,i}, t), \boldsymbol{y}_{t,i})\}_{i=1}^{N_t}$, where $\boldsymbol{x}_{t,i} \in \boldsymbol{X}$ is the input data and $\boldsymbol{y}_{t,i} \in \boldsymbol{Y}_t \in \boldsymbol{Y}$ is its class label. The goal of TIL is to construct a predictor $f$: $\boldsymbol{X} \times \mathbb{T} \to \boldsymbol{Y}$ to predict the class label $y \in \boldsymbol{Y}_t$ for $(\mathbf{x}, t)$ (a given test instance $\mathbf{x}$ from task $t$).

**Knowledge Transfer (KT).** Let $\mathbb{T}_{sim}$ / $\mathbb{T}_{dis}$ be a set of similar/dissimilar tasks of the current task $t$ ($\mathbb{T}_{sim}, \mathbb{T}_{dis} \subseteq \mathbb{T}, \mathbb{T}_{dis} = \mathbb{T} - \mathbb{T}_{sim}$). The learner should transfer the knowledge learned in the past **forward** and leverages it to learn $t$ better, and additionally, the learning of $t$ should also improve the tasks in $\mathbb{T}_{sim}$ (**backward** KT). (Ke et al., 2020) suggested that a TIL model/method should satisfy the basic requirements: (1) overcoming CF and (2) performing FKT and/or BKT to improve the performance of the TIL model across similar tasks.

### 3.1 THEORETICAL ANALYSIS OF THE CF PROBLEM

Given a fixed capacity DNN model with $L$ layers, its weight matrix is denoted by $\mathbb{W} = \{\mathbb{W}^l\}_{l=1}^{L}$, where $\mathbb{W}^l$ is the layer-wise weight matrix for layer $l$. Let $\mathcal{L}(\mathbb{W}_t)$ be the loss function, e.g., cross-entropy loss, where $\mathbb{W}_t$ is the weight matrix of the model after learning task $t \in [1, T]$.

**Input, Gradient, and Weight Spaces.** Given a DNN model, we define the space of the input data as the **Input Space** denoted by $\boldsymbol{S}_{in}$, and the space of the gradient span as the **Gradient Space** $\boldsymbol{S}_g$. We call the space/domain of the weight matrix of a model as its **Weight Space** denoted by $\boldsymbol{S}_w$.

When learning a new task $t$, the weight matrix can be updated by SGD (Amari, 1993) as follows:

$$\mathbb{W}_t = \mathbb{W}_{t-1} - \lambda \nabla \mathbb{W}_t, \mathbb{W}_{t-1}, \mathbb{W}_t \in \boldsymbol{S}_w, \nabla \mathbb{W}_t \in \boldsymbol{S}_g \tag{1}$$

where $\lambda$ is the learning rate and $\nabla \mathbb{W}_t$ is the gradient with respect to $\mathbb{W}_t$ in task $t$ learning. Eq.(1) implies the followings: **(1)** the weight matrix $\mathbb{W}_t$ ($\mathbb{W}_{t-1}$) records the knowledge learned from task $t$ ($t-1$), **(2)** with the quantity of $\lambda \nabla \mathbb{W}_t$, the new task $t$ learning may interfere with the knowledge (stored in $\mathbb{W}_{t-1}$) learned from previous tasks $\leq t-1$, which can cause CF. And with further theoretical analysis, we present and prove Theorem 1 below, which has not been reported in the existing literature. Please see the proof of Theorem 1 in Appendix A.

**Theorem 1.** The learning of the new task $t \leq T$ will not interfere with the knowledge (weights $\boldsymbol{W}_{t-1}$) of the previous learned $t-1$ tasks (i.e., there is no CF) *if and only if* 1) task $t$ occupies its own independent weight subspace $\boldsymbol{W}_t$. that is orthogonal to all other weight subspaces of the previous tasks in the whole weight space $\boldsymbol{S}_w$ of the model, and 2) the modified weights spanned by the orthogonal gradient to the previous learned tasks are imposed in its weight subspace $\boldsymbol{W}_t$ of task $t$.

Theorem 1 shows that when learning task $t$, the weight modification should happen in its own weight subspace $\boldsymbol{W}_t$ to ensure the weight-level orthogonality between tasks rather than only the gradient-level orthogonality. The weight space and gradient space, although dependent, are two different spaces. Thus existing OG-based methods cannot guarantee zero CF as they don't satisfy the second condition in Theorem 1. Moreover, an *orthogonal weight* subspace based (OW-based for short) TIL method following Theorem 1 also yields the benefit stated in Lemma 1 below.

**Lemma 1.** The total number of continually learnable tasks of the OG-based methods is determined by the rank value of its weight matrix in a fixed-size network (Zeng et al., 2019). With the rank value as the metric, the upper bound of the minimum number of learnable tasks of the OW-based TIL method is far greater than that of the OG-based methods (see **Theorems 2-3** and its proof in Appendix A).

## 3.2 THEORETICAL ANALYSIS OF THE KT PROBLEM

We first introduce some definitions and then explore what factors cause positive or negative KT.

**Forward Negative Knowledge Transfer Margin (FNM).** Given two similar tasks $i$ and $t$ ($i < t$) in the TIL setting, let $\epsilon_t$ be the test error of task $t$. $f'(i, t)$ denotes that task $t$ performs its learning with the help of the knowledge of the previous task $i$, and $f'(\emptyset, t)$ otherwise. Then, negative forward knowledge transfer (FKT) happens when $\epsilon_t(f'(i, t)) > \epsilon_t(f'(\emptyset, t))$.

**Backward Negative Knowledge Transfer Margin (BNM).** Given two similar tasks $i$ and $t$ ($i < t$) in the TIL setting, let $\epsilon_i(f'(i))$ be the test error of task $i$ before task $t$ learning, and $\epsilon'_i(f'(i, t))$ be the test error of task $i$ after task $t$ has been learned, then negative backward knowledge transfer (BKT) happens when $\epsilon'_i(f'(i, t)) > \epsilon_i(f'(i))$. Thus, the negative FKT and BKT margins can be defined as:

$$FNM = \epsilon_t(f'(i, t)) - \epsilon_t(f'(\emptyset, t)); BNM = \epsilon'_i(f'(i, t)) - \epsilon_i(f'(i)), i < t, t \in [2, T] \quad (2)$$

**Proposed FNM/BNM-based KT Metrics.** From Eq. (2), it is clear that the degree of forward negative KT/backward negative KT can be evaluated by the FNM/BNM, and that the negative KT occurs if the FNM/BNM is positive. As $\epsilon_t$ is inversely proportional to the test accuracy of task $t$ (denoted by $A_t$) and FNM/BNM may not always be computable, the degree of FWT and BWT across similar tasks $i$ and $t$ in CL can be evaluated by the following Eq.(3)

$$FWT = A_t(f'(i, t)) - A_t(f'(\emptyset, t)); BWT = A'_i(f'(i, t)) - A_i(f'(i)), i < t, t \in [2, T] \quad (3)$$

where $A'_i$ is the test accuracy of task $i$ after task $t$ learning. The greater the positive/negative value of FWT/BWT, the greater the quantity of positive/negative FWT/BWT.

**Theoretical Bound for KT.** Given two similar tasks $i$ and $t$ ($i, t \in [1, T]$) in CL, we now analyze the theoretical bound for KT between similar tasks $i$ and $t$ in their TIL so as to investigate the factors that lead to forward or backward positive/negative KT between them.

Let a *hypothesis* be a mapping function of a neural network for classification of TIL $h : \boldsymbol{X} \times \mathbb{T} \to \boldsymbol{Y}$. According to the test data distribution $\mathcal{D}'_t$ of task $t$, the test error showing that the hypothesis $h$ disagrees with its labeling function $l_t$ (which can also be a hypothesis) is defined as:

$$\epsilon_t(h, l_t) = \mathbb{E}_{\mathbf{x} \sim \mathcal{D}'_t} \left[ |h(\mathbf{x}, t) - l_t(\mathbf{x})| \right] \quad (4)$$

For simplicity, we also denote the *risk* or *error* of hypothesis $h$ on task $t$ by $\epsilon_t(h)$ ($= \epsilon_t(h, l_t)$). With the assumption that the training and the test data are i.i.d (independently identically distributed), we can get its empirical error $\hat{\epsilon}_t(h)$ from the training data. Let the divergence of the test data distributions of $\mathcal{D}'_i$ and $\mathcal{D}'_t$ be $d(\mathcal{D}'_i, \mathcal{D}'_t)$ of tasks $i$ and $t$, where $d(.)$ can be calculated by a similarity/distance metric. Then we can derive and prove the following Theorem 4.

**Theorem 4.** The theoretical bounds for FWT and BWT of tasks $i$ and $t$ are respectively as follows:

$$\epsilon_t(h) \leq \epsilon_i(h) + d(\mathcal{D}'_i, \mathcal{D}'_t) + \min\{\mathbb{E}_{\mathcal{D}'_i}(\mathbb{S}), \mathbb{E}_{\mathcal{D}'_t}(\mathbb{S})\}; \epsilon_i(h) \leq \epsilon_t(h) + d(\mathcal{D}'_i, \mathcal{D}'_t) + \min\{\mathbb{E}_{\mathcal{D}'_i}(\mathbb{S}), \mathbb{E}_{\mathcal{D}'_t}(\mathbb{S})\} \quad (5)$$

where $\mathbb{S} = |l_i(\mathbf{x}) - l_t(\mathbf{x})|$ represents the absolute difference of the test results on text data $\mathbf{x}$ of tasks $i$ and $t$. The proof is given in Appendix A.

From Theorem 1, we observe the followings: (1) In the forward/backward KT process, two additional losses are introduced, (i) the loss due to the divergence of the test data distributions of tasks $i$ and $t$ (the second term on the right side of Eq.(5), and (ii) the loss due to the difference of data classification results of the tasks (the third term). (2) *It is clear that the necessary and sufficient conditions for elimination of negative forward/backward KT is that the errors introduced by the above two terms are zero.* (3) The less the two additional losses above are, the greater the gain of FWT or BWT will be. In practice, however, Eq.(5) may not always be computable as it is impossible to get the test data during model training. Thus, with the assumption that the training and test data are i.i.d, we can employ the empirical errors $\hat{\epsilon}_t(h)$ and $\hat{\epsilon}_i(h)$ to approximate $\epsilon_t(h)$ and $\epsilon_i(h)$ using training data.

Moreover, related KT researches (Zhang et al., 2022; Prado & Riddle, 2022; Wang et al., 2021; Riemer et al., 2019; Lin et al., 2022a;b) have proven the following Theorem 5.

**Theorem 5.** Low similarity or negatively correlated tasks will result in negative KT. Only high similarity tasks or positively correlated tasks can achieve positive KT.

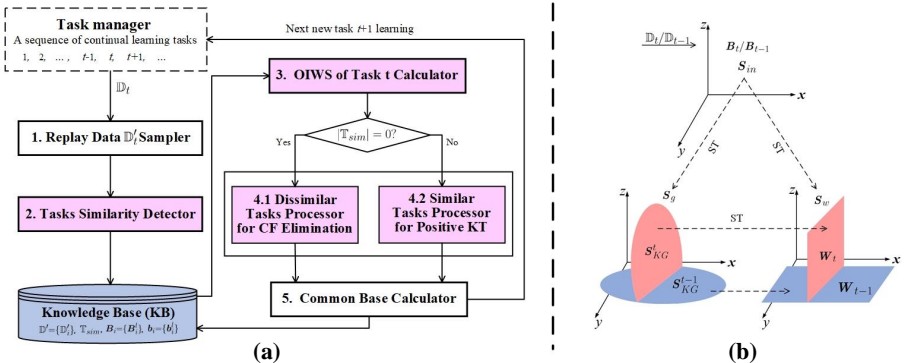

Figure 1: (a) The architecture and pipeline of the proposed STIL, where the proposed new techniques are embedded in the pink components. (b) The space transform (**ST**) diagram of **Input Space** $S_{in}$, **Gradient Space** $S_g$ and **Weight Space** $S_w$. Note that $S_{in}$, $S_g$ and $S_w$ are three related but distinct spaces. With matrix SVD and approximation, the orthogonal common bases $B_t$ and $B_{t-1}$ of significant representations of task $t$ and previous $t-1$ tasks are obtained, by which the orthogonal important gradients $S_{KG}^t$ ($S_{KG}^{t-1}$) and weights $W_t$ ($W_{t-1}$) of task $t$ ($t-1$) can be obtained by ST.

## 4 PROPOSED METHOD STIL

The architecture and pipeline of the proposed STIL method are shown in Figure 1(a) and **Algorithm 3 STIL** is given in Appendix C. STIL's core techniques (pink components) consist of the proposed OIWS-based TIL strategy for maximally CF prevention, the tasks similarity detector, and dissimilar/similar tasks processor (components 4.1 and 4.2) which enable STIL to adaptively switch the focus onto performing KT when the new task is found to be similar to some old tasks or onto overcoming CF when the new task is dissimilar to all old tasks. We discuss the details below.

### 4.1 PROPOSED OIWS OF TASK $t$ AND OIWS-BASED TIL STRATEGY FOR OVERCOMING CF

Motivated by the theoretical study and by the fact that the learned knowledge of task $t$ is determined by its significant/important weights, *we first propose the **Orthogonal Important Weight Subspace (OIWS)** concept for overcoming CF along with fast knowledge indexing and transferring.* The basic ideas are: (1) we let the important weights of each new task occupy an independent subspace in the network weight space $S_w$, called the important weight subspace (IWS). As all the IWSs among tasks are orthogonal to each other, the IWS of a task is known as an OIWS of the task, and (2) we give the exact mapping or indexing relationships between task input space $S_{in}$, gradient space $S_g$ and weight space $S_w$ so as to lay the foundation for reusing the learned knowledge stored in the weight matrix $\mathbb{W}$ (see Figure 1(b)). Specifically, The proposed OWIS-based TIL method consists of two steps:

**Step 1**: Using GPM (Saha et al., 2021) method and dataset $\mathbb{D}_t$ of task $t$, we calculate all the important gradients with large values for task $t$, which is orthogonal to the important gradients of all $t-1$ previous tasks, where $\{b_t^l\}_{l=1}^L$ is the calculated **layer-wise bases of the orthogonal gradients of task** $t$ **at layer** $l$ from input data. As $\{b_t^l\}_{l=1}^L$ is an important piece of knowledge of task $t$, it is stored in the Knowledge Base (KB) of STIL (see Figure 1(a)). Then the **layer-wise common bases** $B_t^l$ of task $t$ and previously $t-1$ tasks are calculated, where $B_1^l = b_1^l$ (see **Algorithm 2** in Appendix C).

**Step 2:** It has two substeps: learning the new task $t$ and preparing for learning the next task $t+1$.

**Step 2.1**: In learning a new task $t$ ($t \in [2, T]$), we restrict the weight updating of task $t$ in its own OIWS to maximally eliminate CF. Note that the OIWS of task $t$ consists of those weights that the important gradients span and update the weights of the model in task $t$ learning, and the OIWS of task $t$ is orthogonal to the other OIWSs of all previous learnt $t-1$ tasks. Thus, by the Space Transformation (ST) method in (Deisenroth et al., 2020), we first obtain the layer-wise OIWS of task $t$ as follows (see Figure 1(b)):

$$W_t^l = W_{t-1}^l - W_{t-1}^l B_{t-1}^l (B_{t-1}^l)^\mathsf{T}, t \in [2, T], l \in [1, L] \tag{6}$$

where $\boldsymbol{W}_t^l$ is the orthogonal projection of $\boldsymbol{W}_{t-1}^l$ onto $\boldsymbol{B}_{t-1}^l$, where $\boldsymbol{B}_{t-1}^l$ is the calculated layer-wise common bases of task $t-1$ after learning task $t-1$. Then we restrict weight updates of task $t$ to its OIWS $\boldsymbol{W}_t^l$, and take its gradients in the directions orthogonal to that of the previous $(t-1)$ tasks.

$$\boldsymbol{W}_t^l = \boldsymbol{W}_t^l - \lambda \nabla_{\boldsymbol{W}_t^l}\mathcal{L}, \quad \nabla_{\boldsymbol{W}_t^l}\mathcal{L} = \nabla_{\boldsymbol{W}_t^l}\mathcal{L} - (\nabla_{\boldsymbol{W}_t^l}\mathcal{L})\boldsymbol{B}_{t-1}^l(\boldsymbol{B}_{t-1}^l)^\intercal, t \in [2, T], l \in [1, L] \quad (7)$$

where $\mathcal{L}$ is the model loss that can take $\mathcal{L}_{dis}/\mathcal{L}_{sim}$ for dissimilar/similar tasks designed specifically to resist CF and/or to achieve KT respectively, and $\nabla_{\boldsymbol{W}_t^l}\mathcal{L}$ is the gradient of the loss with respect to $\boldsymbol{W}_t^l$. Note that the cross entropy loss is employed as the loss function $\mathcal{L}_{dis}$ for dissimilar tasks.

**Step 2.2**: Computing the base $\boldsymbol{b}_t^l$ of the OIWS of task $t$ after learning task $t$ and the common base $\boldsymbol{B}_t^l$ of both $\boldsymbol{B}_{t-1}^l$ and $\boldsymbol{b}_t^l$ to prepare for the next task $(t+1)$ learning (see **Algorithm 2** basesComputing Appendix C). And then iterating Steps 1 and 2 until all learning tasks are completed.

## 4.2 $\mathcal{L}_{sim}$ LOSS WITH CONSTRAINTS AND DATA RELAY TO FACILITATE KT

If there are similar tasks (i.e., $\mathbb{T}_{sim} \neq \emptyset$) for a new task $t$, we want to learn $t$ better by encouraging forward and backward KT from the top-$k$ (we use the empirical value of $k = 2$ to be conservative) previously learned similar tasks in $\mathbb{T}_{sim}$. Note that similar tasks are different tasks with some commonality (i.e., shared knowledge) and also individuality.

Given two similar tasks $i$ and $t$ ($i < t$), we want to achieve positive FKT, i.e., the knowledge learned from $i$ can help learn $t$ better, and to achieve positive BKT, i.e., the learning of task $t$ could improve the accuracy of task $i$. How to achieve the two objectives is a challenge because for the OG-based methods (including the proposed OIWS-based method), each learning task occupies a separate gradient or weight subspace orthogonal to that of the other tasks. Then these methods are inherently incapable of KT unless additional mechanisms are added. To tackle the challenge, inspired by the optimization with constraints, we propose a new entropy-cross loss with constraints $\mathcal{L}_{sim}$.

$$\mathcal{L}_{sim} = -\frac{1}{N}\sum_{i=1}^N y_i log\hat{y}_i, \ s.t. \ \cos(\boldsymbol{W}_t^l \otimes \boldsymbol{W}_{\mathbb{T}_{sim}}^l) = 1, \cos(\boldsymbol{W}_t^l \otimes \boldsymbol{W}_{\mathbb{T}_{dis}}^l) = 0 \quad (8)$$

where $\boldsymbol{W}_{\mathbb{T}_{sim}}^l/\boldsymbol{W}_{\mathbb{T}_{dis}}^l$ is the layer-wise weight matrix of the top-$k$ tasks from $\mathbb{T}_{sim}/\mathbb{T}_{dis}$. $\otimes$ is the dot product of vectors by entries. The two constraints of $\mathcal{L}_{sim}$ try to pull weights of task $t$ to be closer to the top-$k$ similar previous tasks for KT and far away from the top-$k$ dissimilar tasks in $\mathbb{T}_{dis}$ to avoid CF. For $\boldsymbol{W}_t^l$ and $\boldsymbol{W}_{\mathbb{T}_{sim}}^l/\boldsymbol{W}_{\mathbb{T}_{dis}}^l$ in Eq.(8), unlike all existing KT-based methods, STIL requires no additional mechanism or storage and can quickly calculate them using $\boldsymbol{W}_t^l = \mathbb{W}^l \boldsymbol{b}_t^l(\boldsymbol{b}_t^l)^\intercal(\boldsymbol{W}_t^l \in \mathbb{W}^l \in \mathbb{W}, l \in [1, L])$ via retrieving the base $\boldsymbol{b}_t^l$ of task $t$ from its KB. And the similar/dissimilar tasks in $\mathbb{T}_{sim}/\mathbb{T}_{dis}$ can be calculated by following Eq. (11).

Further, to avoid extensive computation in constrained optimization and to accelerate calculation convergence, considering the orthogonality of these two constraints and the contrastive loss, we convert the constraints in Eq. (8) to another objective leading to $\mathcal{L}_{sim}$ becoming the following bi-objective loss.

$$\mathcal{L}_{sim} = -\frac{1}{N}\sum_{i=1}^N y_i log\hat{y}_i - log\frac{1}{N}\sum_{i=1}^N \frac{exp\left(\sum_{j \in \mathbb{T}_{sim}} cos(\boldsymbol{W}_t^l, \boldsymbol{W}_j^l)\right)}{exp\left(\sum_{k \in \mathbb{T}_{sim} \cup \mathbb{T}_{dis}} cos(\boldsymbol{W}_t^l, \boldsymbol{W}_k^l)\right)} \quad (9)$$

**More about knowledge transfer:** The above OIWM-based TIL method may still limit KT as each task occupies its own independent weight subspace. To address the issue, we propose two strategies: (1) using the OWIM' bases of previous learned similar/ dissimilar tasks stored in the KB to quickly index and calculated the knowledge of these tasks in $\boldsymbol{W}_t^l$ and $\boldsymbol{W}_{\mathbb{T}_{sim}}^l/\boldsymbol{W}_{\mathbb{T}_{dis}}^l$ (see Eq. (8) for reuse or share of the knowledge, and (2) using **replay data**[2] $\mathbb{D}'$ in KB (see the next subsection) to transfer the shared knowledge among similar tasks to further encourage forward KT. That is, when learning a new task $t$, we add the replay data of the previous tasks that are most similar to task $t$ to the training set of $t$ for joint training. With the two strategies, STIL performs its positive FKT. By selecting those of the similar tasks that would not cause negative BKT calculated by Eqs. (5) and (3), STIL achieves its positive BKT. Refer to **Algorithm 3 STIL** given in Appendix C.

---

[2] The replay data $\mathbb{D}'$ are extracted and accumulated online, i.e., when a new task $t$ comes, STIL extracts a subset $\mathbb{D}_t'$ from the training dataset $\mathbb{D}_t$ of $t$ with a random sampling rate of 5% for each class, and then adds $\mathbb{D}_t'$ to the replay data $\mathbb{D}'$.

### 4.3 SIMILARITY/DISSIMILARITY DETECTION METHOD

Theorems 4 and 5 indicate that (1) an accurate measure of task similarity is essential for positive KT, and (2) to ensure positive KT, the divergence of data distributions and the difference of the data classification results must be considered together. We note that after the model has learned $t-1$ tasks, all the knowledge learned is recorded in the weight matrix $\mathbb{W}_{t-1}$ of the model. We also observed in experiments that for a new task $t$ learning, if $t$ gains improved performance on $\mathbb{W}_{t-1}$ compared with on $\mathbb{W}$ with randomly initialized weights and no training, it indicates there is shared knowledge in $\mathbb{W}_{t-1}$ for task $t$, i.e., there must be similar previous tasks to task $t$; Otherwise, there is no similar task.

Based on the above observations, we now propose a method to detect task similarity based only on the current model. Given a model denoted by $model_{ori}$ (with the same architecture as the STIL model, which is denoted by $model_{CL}$) with randomly initialized weights and no training, and the replay data $\mathbb{D}' = \{\mathbb{D}'_1, ..., \mathbb{D}'_{t-1}, \mathbb{D}'_t\}$. STIL does task similarity detection as follows:

**Step 1:** Feeding $\mathbb{D}'$ into $model_{ori}$ sequentially to obtain their corresponding original loss distributions $L' = \{L'_1, ..., L'_{t-1}, L'_t\}$. **Step 2:** Before starting to learn a new task $t$, feeding $\mathbb{D}'_1, ..., \mathbb{D}'_t$ into $model_{CL}$ sequentially to get its loss distributions $L = \{L_1, ..., L_{t-1}, L_t\}$ from $model_{CL}$. **Step 3:** Calculating the distances between tasks $i$ and $t$ with respect to the distributions $L'$ and $L$.

$$dis' = dis(L_i', L_t')/\sum_{i=1}^{t-1} dis(L_i', L_t') \, , \, dis = dis(L_i, L_t)/\sum_{i=1}^{t-1} dis(L_i, L_t), i \in [1, t-1] \quad (10)$$

where $dis'$ represents the original/true Loss Distribution Distance (LDD) of the two tasks as there is no knowledge of any task in $model_{ori}$, and $dis$ denotes the LDD of the two tasks based on some learned knowledge in $model_{CL}$.

Based on the above observations and Eq. (10), we can infer that if $dis < dis'$, it indicates that task $i$ and task $t$ have some shared knowledge in $model_{CL}$, i.e., they are some similar tasks, so their LDD is going to be closer than their initial LDD value $dis'$, and vice versa. Thus, we propose a simple **Similarity or Dissimilarity Metric (SDM)** to measure the similarity/dissimilarity of tasks $i$ and $t$.

$$SDM = \begin{cases} dis > dis' \, , \, |dis - dis'| < \theta & i, t \in \mathbb{T}_{dis} \\ dis < dis' \, , \, |dis - dis'| \geq \theta & i, t \in \mathbb{T}_{sim} \end{cases} , \quad i < t, t \in [2, T] \quad (11)$$

where $\theta$ is a distance threshold. Based on the theoretical research for KT (Eq. (5) and Theorems 4 and 5), in this paper, $\theta$ takes an empirical value greater than or equal to 0.5 to ensure positive KT.

**Step 4:** Calculating the layer-wise similarity between tasks $i$ (an old task) and $t$ by Eq. (11). Please refer to **Algorithm 1** LWSimilarity in Appendix C.1 for details.

Following Theorems 4 and 5, and considering the good properties of Wasserstein distance (Vallender, 1974; van den Oord et al., 2018) (see Figure 7 in Appendix): when calculating the distance of the data, 1) it has no assumptions on the distribution of the data and does not need to know the type of the distribution; and 2) it takes into account not only the distance, but also the shape/geometry of the data, we employ the Wasserstein distance in the calculation of Eq.(11).

## 5 EXPERIMENTAL RESULTS

### 5.1 EXPERIMENT SETUP

**Datasets (dissimilar tasks).** For this set of experiments, we use five benchmark image classification datasets: (1.1) PMNIST (10 tasks), (1.2) CIFAR 100 (10 tasks), (1.3) CIFAR 100 Sup (20 tasks), (1.4) MiniImageNet (20 tasks), and (1.5) 5-Datasets (5 tasks). We regard the tasks in each dataset as dissimilar as each task has different/disjoint classes. Note that two datasets CIFAR 100 Sup and 5-Datasets are datasets with "difficult" tasks (Saha et al., 2021).

**Datasets (similar tasks).** (2.1) F-EMNIST-1 (10 tasks), (2.2) F-EMNIST-2 (35 tasks), (2.3) F-CelebA-1 (10 tasks), and (2.4) F-CelebA-2 (20 tasks). We consider tasks in F-EMNIST and F-CelebA are similar as each task in F-EMNIST contains one writer's written digits/characters and each task in F-CelebA contains images of one celebrity labeled by whether he/she is smiling or not.

**Datasets (mixed tasks).** (3.1) (EMNIST, F-EMNIST-1) (20 tasks) and (3.2) (CIFAR 100, F-CelebA-1) (20 tasks). Each of them is a combination of the task sequence from F-EMNIST-1 (or F-CelebA-1) and the dissimilar task sequence EMNIST (or CIFAR 100) with the tasks randomly mixed.

Table 1: ACC and BWT performances with standard deviations over 5 different runs of the proposed ETCL and 14 strong baselines of the 4 categories on five dissimilar benchmark datasets.

| Datasets | | PMNIST (10 Tasks) | | CIFAR 100 (10 Tasks) | | CIFAR 100 Sup (20 Tasks) | | MiniImageNet (20 Tasks) | | 5-Datasets (5 Tasks) | | Average | |
|---|---|---|---|---|---|---|---|---|---|---|---|---|---|
| Type | Methods | ACC(%) | BWT | ACC(%) | BWT | ACC(%) | BWT | ACC(%) | BWT | ACC(%) | BWT | ACC(%) | BWT |
| | **ONE** | **96.70** | None | **79.58** | None | **61.00** | None | **69.46** | None | **93.58** | None | **80.06** | None |
| (1) | LwF | $85.72_{\pm0.47}$ | $-0.11_{\pm0.01}$ | $67.70_{\pm0.37}$ | $-0.08_{\pm0.01}$ | $51.55_{\pm0.49}$ | $-0.03_{\pm0.01}$ | $60.51_{\pm0.32}$ | $-0.03_{\pm0.01}$ | $89.10_{\pm0.57}$ | $-0.02_{\pm0.01}$ | 70.92 | -0.05 |
| | DEN | $91.17_{\pm0.49}$ | $-0.03_{\pm0.01}$ | $68.84_{\pm0.25}$ | $-0.03_{\pm0.01}$ | $51.10_{\pm0.41}$ | $-0.03_{\pm0.01}$ | $56.58_{\pm0.42}$ | $-0.04_{\pm0.01}$ | $79.75_{\pm0.53}$ | $-0.01_{\pm0.01}$ | 69.49 | -0.03 |
| | APD | $92.48_{\pm0.59}$ | $-0.03_{\pm0.01}$ | $72.49_{\pm0.43}$ | $-0.03_{\pm0.01}$ | $56.81_{\pm0.44}$ | $-0.02_{\pm0.01}$ | $58.73_{\pm0.51}$ | $-0.03_{\pm0.01}$ | $83.72_{\pm0.54}$ | $-0.07_{\pm0.01}$ | 72.86 | -0.04 |
| (2.1) | A-GEM | $83.56_{\pm0.16}$ | $-0.13_{\pm0.01}$ | $63.98_{\pm1.22}$ | $-0.15_{\pm0.02}$ | $42.78_{\pm0.89}$ | $-0.13_{\pm0.05}$ | $57.24_{\pm0.72}$ | $-0.12_{\pm0.01}$ | $84.04_{\pm0.33}$ | $-0.12_{\pm0.01}$ | 66.33 | -0.13 |
| | OWM | $90.71_{\pm0.11}$ | $-0.02_{\pm0.01}$ | $50.94_{\pm0.60}$ | $-0.03_{\pm0.01}$ | – | – | – | – | – | – | 70.83 | -0.03 |
| | OGD | $82.50_{\pm0.13}$ | $-0.14_{\pm0.01}$ | $47.12_{\pm0.87}$ | $-0.04_{\pm0.01}$ | $36.92_{\pm0.57}$ | $-0.03_{\pm0.04}$ | $44.89_{\pm0.49}$ | $-0.04_{\pm0.02}$ | $57.12_{\pm0.41}$ | $-0.04_{\pm0.01}$ | 53.71 | -0.06 |
| | GPM | $93.91_{\pm0.16}$ | $-0.03_{\pm0.01}$ | $72.48_{\pm0.40}$ | $-0.03_{\pm0.01}$ | $57.10_{\pm0.38}$ | $-0.03_{\pm0.01}$ | $60.41_{\pm0.01}$ | $-0.03_{\pm0.04}$ | $91.22_{\pm0.22}$ | $-0.01_{\pm0.00}$ | 75.02 | -0.03 |
| | EWC | $89.97_{\pm0.57}$ | $-0.04_{\pm0.01}$ | $68.80_{\pm0.88}$ | $-0.02_{\pm0.01}$ | $41.49_{\pm0.79}$ | $-0.03_{\pm0.02}$ | $52.01_{\pm2.53}$ | $-0.12_{\pm0.03}$ | $86.61_{\pm0.20}$ | $-0.05_{\pm0.01}$ | 64.18 | -0.05 |
| | UCL | $89.53_{\pm0.22}$ | $-0.05_{\pm0.01}$ | $64.08_{\pm0.46}$ | $-0.06_{\pm0.02}$ | $47.22_{\pm0.53}$ | $-0.09_{\pm0.02}$ | $45.85_{\pm0.41}$ | $-0.10_{\pm0.01}$ | $88.54_{\pm0.38}$ | $-0.05_{\pm0.02}$ | 67.04 | -0.07 |
| (2.2) | HAT | $90.35_{\pm0.32}$ | **$-0.00_{\pm0.00}$** | $72.06_{\pm0.30}$ | $-0.00_{\pm0.00}$ | $55.85_{\pm0.37}$ | **$-0.00_{\pm0.00}$** | $59.78_{\pm0.47}$ | $-0.03_{\pm0.01}$ | $91.32_{\pm0.18}$ | $-0.01_{\pm0.00}$ | 73.87 | -0.01 |
| | SupSup | $96.03_{\pm0.12}$ | **$-0.00_{\pm0.00}$** | $74.63_{\pm0.36}$ | $-0.00_{\pm0.00}$ | **$61.53_{\pm0.23}$** | **$-0.00_{\pm0.00}$** | $61.55_{\pm0.20}$ | **$-0.00_{\pm0.00}$** | **$92.30_{\pm0.19}$** | $-0.00_{\pm0.00}$ | 77.21 | **-0.00** |
| (2.3) | CAT | $93.87_{\pm0.51}$ | $-0.03_{\pm0.01}$ | $59.06_{\pm0.49}$ | $-0.08_{\pm0.01}$ | $50.23_{\pm0.32}$ | $-0.02_{\pm0.01}$ | $59.55_{\pm0.61}$ | $-0.03_{\pm0.01}$ | $86.05_{\pm0.74}$ | $-0.04_{\pm0.03}$ | 69.75 | -0.04 |
| | TRGP | $96.34_{\pm0.11}$ | $-0.08_{\pm0.01}$ | $73.95_{\pm0.32}$ | $-0.02_{\pm0.01}$ | $58.48_{\pm0.01}$ | $-0.01_{\pm0.00}$ | $60.73_{\pm0.60}$ | $-0.02_{\pm0.06}$ | $92.22_{\pm0.10}$ | $-0.04_{\pm0.01}$ | 76.47 | -0.03 |
| | CUBER | **$97.04_{\pm0.11}$** | $-0.02_{\pm0.01}$ | $74.67_{\pm0.22}$ | **$0.01_{\pm0.00}$** | $57.92_{\pm0.01}$ | $-0.01_{\pm0.00}$ | $62.67_{\pm0.35}$ | $-0.01_{\pm0.04}$ | $91.95_{\pm0.30}$ | **$0.03_{\pm0.00}$** | 77.70 | **-0.00** |
| | STIL(**Ours**) | $97.15_{\pm0.03}$ | $-0.02_{\pm0.01}$ | $75.28_{\pm0.13}$ | $0.02_{\pm0.00}$ | $63.78_{\pm0.01}$ | $-0.01_{\pm0.00}$ | $65.11_{\pm0.11}$ | **$-0.00_{\pm0.01}$** | $93.46_{\pm0.06}$ | $-0.01_{\pm0.01}$ | 78.36 | -0.00 |

As **ONE** (see footnote 3 on this page) has no knowledge transfer and no forgetting involved, its BWT and FWT are denoted by **None**. As CAT is bound to its specific network structure, its experimental results are run according to its network structure and source code. Other methods use the same backbone network on each dataset shown in Appendix D. "–" indicates that the source codes are not provided by the baselines leading to no experimental results. The blue results mean the best prior results.

**Baselines.** We compare STIL with 14 state-of-the-art baselines of 4 categories: (1) *Expansion-based methods*: LwF, DEN and APD; (2) *Non-Expansion method* (2.1) **Experience-replay/OG/Regularization** based methods: A-GEM, OWM, OGD, GPM, EWC and UCL; (2.2) **Parameter isolation** based methods: HAT and SupSup; (2.3) **KT-based** methods. CAT, TRGP and CUBER. We use the official codes for these baselines.

Refer to Appendix B for the details about these datasets, baselines, and implementations.

**Performance Metrics.** (1) **Average accuracy (ACC)** of all tasks after the last task has been learned: $ACC = \frac{1}{T}\sum_{i=1}^{T} R_{T,i}$. (2) **Backward transfer (BWT)**: $BWT = \frac{1}{T-1}\sum_{i=1}^{T-1}(R_{T,i} - R_{i,i})$ (Lopez-Paz & Ranzato, 2017), which indicates how much the new task affect the old tasks, the greater the positive value, the better. Here, $T$ is the total number of tasks and $R_{T,i}$ is the accuracy of the model on $i^{th}$ task after learning the last task $T$. A negative value (i.e., *forgetting rate*) indicates CF and a positive value indicates positive BKT. (3) **Forward transfer (FWT)**: Following Eq. (3) $FWT = \frac{1}{T-1}\sum_{i,t}(A_t(f(i,t)) - A_t(f(\emptyset,t)))$, where $A_t(f(\emptyset,t))$ can be calculated by method ONE [3] $(i < t, i \in [1, T-1], t \in [2, T])$.

## 5.2 MAIN EXPERIMENTAL RESULTS AND ANALYSIS

**Results of Dissimilar Tasks - Overcoming CF.** The task sequences here consists of only dissimilar tasks, which have little shared knowledge to transfer. We use ACC and BWT (forgetting rate) as the metrics to evaluate CF prevention. Table 1 reports the results, which indicate that STIL outperforms all baselines in ACC , and also has the best average BWT. Compared with the best ACC results of the baselines, STIL achieves the average gains of 0.66% over baselines on the five datasets (or task sequences) (see the rightmost column in Table 1). In particular, STIL can even obtain the closest ACC performance to ONE (see footnote 3 on this page) compared to all baselines on two "difficult" tasks datasets: CIFAR 100 Sup (20 tasks) and 5-Datasets (5 datasets used as 5 tasks), respectively. Although CUBER has good BWT on some datasets, it is not difficult to find that its BWT oscillates between positive and negative values. All experiments show STIL's strong ability to overcome CF.

**Results of Similar Tasks - Knowledge Transfer (KT).** Similar task sequences contain more shared knowledge to transfer. Table 2 reports the FWT and BWT performances of the proposed STIL and 3 strong baselines that were designed for knowledge transfer, i.e., CAT, TRGP and CUBER. We also included GPM as TRGP and CUBER were based on GPM. Table 2 shows that STIL achieves all positive knowledge transfer (FWT and BWT) in four similar tasks datasets resulting in 11.89%, 13.24%, 12.50% and 15.00% ACC gains respectively compared to ONE. As GPM has no explicit KT mechanism and OIWS-based strategy for CF prevention compared with our STIL, its average

---

[3]ONE (one task learning) – building a model for each task independently using a separate neural network same as the model backbone, which has no knowledge transfer and no forgetting involved.

Table 2: FWT (%) and BWT (%) performances with standard deviations of the proposed STIL and 4 strong baselines with/without KT capacity over 5 different runs on four similar task datasets.

| Dataset | F-EMNIST-1 (10 Tasks) | | | F-EMNIST-2 (35 Tasks) | | | F-CelebA-1 (10 Tasks) | | | F-CelebA-2 (20 Tasks) | | | Average | | |
|---|---|---|---|---|---|---|---|---|---|---|---|---|---|---|---|
| Method | ACC (%) | FWT | BWT | ACC (%) | FWT | BWT | ACC (%) | FWT | BWT | ACC (%) | FWT | BWT | ACC (%) | FWT | BWT |
| **ONE** | **69.85** | None | None | **71.55** | None | None | **77.50** | None | None | **73.75** | None | None | **73.16** | None | None |
| **GPM** | $75.18_{\pm 0.06}$ | 3.72 | 2.18 | $79.20_{\pm 0.40}$ | **7.82** | -0.07 | $86.34_{\pm 0.36}$ | **7.41** | 1.04 | $75.05_{\pm 0.30}$ | **1.76** | -0.46 | **78.94** | **5.18** | 0.67 |
| **CAT** | $61.90_{\pm 0.21}$ | -10.41 | 2.59 | $63.00_{\pm 0.25}$ | -9.64 | 1.64 | $76.90_{\pm 0.21}$ | -1.13 | -1.00 | $65.87_{\pm 0.12}$ | -7.88 | 0.00 | 66.92 | -7.27 | 0.81 |
| **TRGP** | $76.66_{\pm 0.46}$ | 4.69 | 3.01 | $79.54_{\pm 0.42}$ | 7.15 | 1.00 | $78.34_{\pm 0.49}$ | 0.75 | 0.00 | $70.24_{\pm 0.35}$ | -3.51 | 0.00 | 76.20 | 2.27 | 1.00 |
| **CUBER** | $78.48_{\pm 0.47}$ | **7.03** | 2.15 | $76.80_{\pm 0.53}$ | 5.78 | -1.26 | $78.35_{\pm 0.55}$ | 0.76 | 0.00 | $70.25_{\pm 0.33}$ | -3.50 | 0.00 | 75.97 | 2.52 | 0.22 |
| **STIL(Ours)** | $81.74_{\pm 0.23}$ | **8.27** | **4.94** | $84.79_{\pm 0.17}$ | **8.35** | **3.89** | $90.00_{\pm 0.11}$ | 3.70 | **10.19** | $88.75_{\pm 0.12}$ | 0.44 | **14.48** | **86.32** | **5.19** | **8.53** |

The backbone ResNet-18 is used for the two similar tasks datasets F-EMNIST. The backbone 3-Layer FCN is used in the next two similar tasks datasets F-CelebA for all algorithms as the two datasets have a small number of samples and ResNet performed poorly. As CAT is bound to its specific network structure 3-Layer FCN, its experimental results are run according to its network structure and source code. The blue results mean the best prior results.

performances (ACC, FWT and BWT) are inferior to those of our STIL. CAT is weak as it only works with 3-Layer FCN. Our STIL is strong in both forward and backward transfer. The average results in the rightmost column show that the STIL is significantly better than the baselines.

**Results for Mixed Tasks- CF Prevention and KT**. Due to space limitations, we put the results for this set of experiments in Table 11 in Appendix D.4. The table shows that STIL still achieves all positive KT (including FKT and BKT) and outperforms all baselines in accuracy.

**Capacity of OG-based methods and our OIWS-based method**. Figure 7 in Appendix D.5 shows that as the number of tasks increases, the existing representative OG-based method OWM suffers more and more from CF as there are no new orthogonal gradient and weight subspace to learn new tasks, which is as revealed by Theorems 2-3 and Lemma 1. STIL's performances are quite stable.

**Efficiency and Memory Performance Comparisons.** The results given in Table 5, Appendix D.1 show that STIL is the best in both time (except GPM) and memory efficiency compared with the baselines OWM, GPM, TRGP and CUBER, and CAT. As OWM has a large number of high-dimensional matrix operations, it not only suffers from high time and space complexity but also cannot be applied to complex DNNs. As CAT, TRGP and CUBER also require more complex operations and larger memory size for their KT, their time and memory requirements are much higher than those of STIL. The running times (the space required) of these baselines are 7.95, 1.33, 2.61, 1.63 and 2.26 (6.05, 1.13, 2.24, 1.66 and 1.92) times on average higher than that of STIL, respectively.

**Ablation Experiments.** The ablation experimental results are given in Table 3. "STIL(-SDM)" denotes without using SDM task similarity metric but using Euclidean distance, "STIL(-OIWS)" means without deploying the OIWS-based TIL strategy in STIL, and "STIL(-KT)" means removing the KT function in STIL. Ablation results show that the full STIL always gives the best ACC and every component, i.e., SDM, OIWS or KT, contribute to the model's per-

Table 3: Ablation experiments of the proposed STIL.

| Datasets | STIL(-SDM) ACC(%) | STIL(-OIWS) ACC(%) | STIL(-KT) ACC(%) | STIL ACC(%) |
|---|---|---|---|---|
| (1.1) | 95.13 | 95.15 | 95.15 | **97.15** |
| (1.2) | 74.33 | 71.00 | 73.65 | **75.28** |
| (1.3) | 56.14 | 57.10 | 58.58 | **60.78** |
| (1.4) | 64.48 | 60.15 | 60.13 | **65.11** |
| (1.5) | 90.96 | 92.35 | 90.10 | **93.46** |
| (2.1) | 75.64 | 80.24 | 71.35 | **81.74** |
| (2.2) | 74.93 | 83.25 | 73.07 | **84.79** |
| (2.3) | 85.83 | 88.69 | 79.50 | **90.00** |
| (2.4) | 80.82 | 87.30 | 75.25 | **88.75** |
| (3.1) | 76.27 | 76.49 | 76.94 | **78.01** |
| (3.2) | 63.25 | 63.96 | 64.46 | **65.46** |

formance. Particularly, on 4 similar datasets, i.e., (2.1) EMNIST-1, (2.2) EMNIST-2, (2.3) F-CelebA-1, and (2.4) F-CelebA-2, if the SDM or KT mechanism is removed from STIL, the accuracy of STIL will drop sharply, which shows the effectiveness of the proposed SDM and KT.

# 6 CONCLUSION

This research first theoretically showed that the existing OG-based methods can only mitigates CF but cannot eliminate CF. As knowledge transfer (KT) is also a core objective of CL but limited research has been done, we theoretically study the KT problem and give the bounds that can lead to negative forward and backward KT. With the goal to avoid CF and perform KT, we proposed a novel task-incremental learning (TIL) method STIL based on weight-level orthogonality between tasks and a new KT mechanism. Extensive experimental results showed that STIL not only self-adaptively performs similar, dissimilar or mixed tasks well in terms of forgetting prevention and knowledge transfer, but also outperforms strong baselines. Our future work will further improve STIL's accuracy.

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
