# OpenReview forum: "Continual Learning with Orthogonal Weights and Knowledge Transfer"
_ICLR.cc/2024/Conference — Submitted to ICLR 2024_

### Official Review · Reviewer_NdAy · 2023-10-28

**Soundness:** 3 good
**Presentation:** 1 poor
**Contribution:** 3 good
**Rating:** 6
**Confidence:** 3

**Summary:**

This work tackles two problems in continual learning: catastrophic forgetting and knowledge transfer. For the first problem, the authors identify that traditional orthogonal gradient based methods cannot fully solve the problem, so they propose to use orthogonal weight space. For the second problem, the authors propose a new loss with constraint to encourage weight similarity for similar tasks. The algorithm achieves good empirical results.

**Strengths:**

1. The work presents a theoretically grounded analysis of the problem of CF and KT. The authors identify a key advantage of orthogonal weight compared to orthogonal gradients, which is novel.

2. The empirical performance is convincing, where the proposed algorithm consistently outperforms baseline. The ablation study demonstrates the effectiveness of both the CF and KT parts.

**Weaknesses:**

1. Problem (2) in the introduction section is questionable for OG. It is unclear what are the properties of important weight and important gradients, which the authors do not provide sufficient explanation. The authors also do not clearly explain the problem of overconsumption of network capacity, and why the performance drop in Figure 10 comes from this reason.

2. I do not think it is a good practice to organize the paper such that important information such as pseudocode for the main algorithm is deferred to the appendix, especially in section 4 and 5. This problem is severe enough such that the text is very difficult to follow without checking the reference. Key experimental results are not shown in the main text. This essentially gains unfair advantage to exceed the page limit.

**Questions:**

1. In the paragraph following Eq.1, what is the notation “/” mean? What are $W_t/W_{t-1}$ and $t/t-1$?

2. Is there a way to measure how close the weights are orthogonal to each other? The ablation experiments do not indicate whether using OIWS indeed improves the performance by achieving orthogonality.

---

> ### Author Response · Authors · 2023-11-23
>
> Dear reviewer NdAy,
>
> Thanks for your valuable comments!
>
> **Q1**: Problem (2) in the introduction section is questionable for OG. It is unclear what are the properties of important weight and important gradients, which the authors do not provide sufficient explanation. The authors also do not clearly explain the problem of overconsumption of network capacity, and why the performance drop in Figure 10 comes from this reason.
>
> **Response**: The important weights and gradients are the weights and gradients with larger weights and gradients that determine the accuracy of a learning task. We clarified the concepts and provided the explanation for Figure 10 (corresponding to Figure 6 in the revised paper). Please see our resubmitted/revised paper.
>
> **Q2**: I do not think it is a good practice to organize the paper such that important information such as pseudocode for the main algorithm is deferred to the appendix, especially in section 4 and 5. This problem is severe enough such that the text is very difficult to follow without checking the reference. Key experimental results are not shown in the main text. This essentially gains unfair advantage to exceed the page limit.
>
> **Response**: Many thanks for your valuable comments! In our revised paper, we have put some important content into the main text. For example, Figure 1 (a)-(b) (Sec.4), Theorem 5 (Sec.3.2), ablation experiments of task similarity metric SDM (embedded in Table 3), etc., and have discussed them more to facilitate understanding.
>
> **Q3**: In the paragraph following Eq.1, what is the notation “/” mean? What $\mathbf{W}_t$/ $\mathbf{W}_{t-1}$ and $t$/$t-1$?
>
> **Response**: The notation "/" means 'or'. Thus $W_t$/$W_{t-1}$ and $t$/$t-1$ represent that task $t$ or $t-1$ corresponds to its OIWS' weights $W_t$ or $W_{t-1}$ of the model.
>
> **Q4**: Is there a way to measure how close the weights are orthogonal to each other? The ablation experiments do not indicate whether using OIWS indeed improves the performance by achieving orthogonality.
>
> **Response**: Let the OWIS-based weights of the learning task $t-1$ and $t$ be $W_{t-1}$ and $W_t$ respectively, then the dot product of $W_{t-1}$ and $W_t$ can represent the degree of orthogonality between them. If they are orthogonal to each other, the dot product value is zero; the smaller the positive value is, the closer they are; the larger the negative value is, the greater the negative correlation between them. In our ablation experiments (see Table 3), it is clearly shown that without deploying the OIWS-based TIL strategy in STIL (denoted by STIL(-OIWS)), on all test datasets, the accuracy (ACC\%) of STIL decreases, which indicates that some forgetting has occurred. Therefore, the OIWS-based strategy plays an important role in STIL's resistance to forgetting.

---

### Official Review · Reviewer_1mbh · 2023-10-28

**Soundness:** 2 fair
**Presentation:** 3 good
**Contribution:** 2 fair
**Rating:** 3
**Confidence:** 5

**Summary:**

This work proposed  weight orthogonal projection for task incremental learning. It provided theoretical analysis of catastrophic forgetting (CF) and knowledge transfer (KT)  and derived the bounds of negative forward and backward KT.  Then,  the authors proposed STIL based on weight orthogonal projection.  STIL introduces orthogonal important weight subspace (OIWS) to avoid CF, which is the key difference compared to prior gradient orthogonal works. To enhance KT, the authors further transfer the shared knowledge among similar prior tasks to the current task, where the task similarity is determined by loss distribution distance replying to data reply.

The paper presented comprehensive experiments.

**Strengths:**

This paper studied a very important problem in continual learning, aiming to eliminate catastrophic forgetting.

+ It proposed a solution   to avoid catastrophic forgetting and improve knowledge transfer, respectively.

+ It conducted comprehensive experiments in various settings and benchmarks, and consistently show better results.

**Weaknesses:**

It seems there is confusion about some fundamental concepts and there are some pitfalls with the proposed approach. Specifically, for each task, the model lies in some (sub)space. In the literature, the orthogonal projection method is proposed to update the model weights and learn the model for each task accordingly. This paper proposed gradient space, which  does not make much sense. Gradient descent is used to update weights in the model space, so what does gradient space refer to?  The gradient in each step is computed using input data and is stochastic; so would gradient space would be random?

This paper claims the proposed weight orthogonal method can eliminate catastrophic forgetting completely; this is overly ambitious.
For instance, when backward knowledge transfer takes place, it would change the weights for previous tasks. How to ensure the weight orthogonality all the way?

Some more detailed comments:

1. For Theorem 1, it’s not  clear why prior OG-based methods cannot guarantee zero CF.  What if the modified weights spanned by the orthogonal gradient to the previously learned tasks  happen on the whole weight space, instead of its own weight subspace?
2. The proposed STIL seems complex: It has two steps for learning each new task with four steps to calculate the similarity between tasks.  Thus the memory and time cost need to be justified. For example,  since the proposed STIL adopts GPM method to obtain knowledge bases with additional computation and memory cost (e.g., data replay), why the proposed STIL can have less time and memory compared to GPM for some datasets (e.g., CIFAR100 Sup, 5-dataset as shown in Table 6)?  How do the authors define memory in this work? Does it include both stored bases and replay data?
3. The experimental results compared to CUBER need to be justified.  As shown in Table.1, compared to CUBER, STIL shows better accuracy but higher forgetting which is inconsistent with the claims.
4. The proposed method seems only can be used to Task-incremental learning. How about class-incremental learning?

**Questions:**

Two additional questions regarding the detailed techniques:

1. The definition of f(slashed zero, t): 1) the function f(slashed zero, t) is defined as “when learning t, it does not use the knowledge of any task”. Does the “any task” mean any prior tasks? 2) How do the authors calculate the accuracy of f(slashed zero, t) as shown in Eq(4).
2. When discussing task similarities in Section 4.2, the authors consider top-k (i.e. k = 2) prior tasks. Will the k values affect the performance for different datasets?

---

> ### Author Response · Authors · 2023-11-23
>
> Dear reviewer 1mbh,
>
> Thanks for your valuable comments!
>
> **Q1**: What does gradient space refer to?
>
> **Response**: Given a DNN model, we define the space of the input data as the Input Space denoted by $\mathbf{S}_{in}$, and the space of the gradient span as the Gradient Space $\mathbf{S}_g$. Please refer to the gradient space definition shown in paragraph 2, Section 3.1.
>
> **Q2**: The gradient in each step is computed using input data and is stochastic; so would gradient space would be random?
>
> **Response**: In the SGD algorithm used in deep learning, ``random'' means that it **randomly extract a batch of samples** from the training dataset of a learning task for model training. Since the data from the training dataset for each task are i.i.d (independently and identical distributed), their gradient space would not be random but come from the same gradient space.
>
> **Q3**: ... How to ensure the weight orthogonality all the way?
>
> **Response**: Yes, you are right. However, in the backward knowledge transfer, the proposed STIL adopts the constraints in Eq.(8) to carry out both forward/backward knowledge transfer and forgetting prevention. Although we cannot guarantee zero forgetting in backward knowledge transfer, the constraint tries to maximally reduce it.
>
> **Q4**: For Theorem 1, it’s not clear ...
>
> **Response**: Section 3.1 has shown that the gradient space and the weight space of the model are two different spaces, although they are related. Therefore, modifying the weights in a gradient orthogonal manner does not guarantee that the modified weights of the current learning task are orthogonal to the weights of the previously learned task, as proven in Theorem 1. Only the weight orthogonality can ensure no CF.
>
> **Q5**: ... why the proposed STIL can have less time and memory compared to GPM for some datasets? ...
>
> **Response**: The reason why time and space are better than GPM on the datasets you mentioned above is due to the optimization we made in the implementation of the system. GPM loads in all its stored information during its computation, while our STIL loads only the information needed for computation on demand. In addition, during model training, we added the processing of early stop of model convergence, that is, once the convergence of model training/learning is determined, the training is immediately ended, unlike GPM, which always needs to be iteratively completed, which have been explained in our revised Appendix D.2.
>
> **Q6**: The experimental results compared to CUBER need to be justified...
>
> **Response**: Although CUBER achieves positive BWT with its good BKT on some datasets, it is not difficult to see that its BWT oscillates between positive and negative values. And more importantly, when learning a new task, not when backward knowledge transfer, its CF prevention and FKT were inferior to our STIL, which results in its performance was not as good as ours in the end. Please see the detailed experimental results (tasks 3-5's ACC performance) shown in Figure 8 in Appendix.
>
> **Q7**: The proposed method seems only can be used to Task-incremental learning. How about class-incremental learning?
>
> **Response**: We believe that the proposed method STIL can be applied to class-incremental learning (CIL) in those CIL methods that use a task-incremental learning method to fully overcome forgetting and a task label prediction to identify the correct task for each test sample.
>
> **Q8**: The definition of $f (\emptyset, t)$: 1) the function $f (\emptyset, t)$ is defined as "when learning t, it does not use the knowledge of any task". Does the "any task" mean any prior tasks? 2) How do the authors calculate the accuracy of $f (\emptyset, t)$ as shown in Eq.(4).
>
> **Response**: 1) "any task" means any prior similar task. 2) For the $f (\emptyset, t)$ calculation in Eq.(4), we built a separate model for the task using a duplicated network with randomized initial parameters and no training, which clearly has no knowledge transfer and no forgetting involved. Please refer to footnote 3 in this paper.
>
> **Q9**: When discussing task similarities in Section 4.2, the authors consider top-k (i.e. k = 2) prior tasks. Will the $k$ values affect the performance for different datasets?
>
> **Response**: Our experimental results show that $k =2$ is good for all datasets. That is why we used the same $k=2$ in all our experiments.

---

### Official Review · Reviewer_c582 · 2023-10-30

**Soundness:** 3 good
**Presentation:** 3 good
**Contribution:** 2 fair
**Rating:** 6
**Confidence:** 5

**Summary:**

The paper proposes a theoretical analysis of the shortcomings of the OG method and proposes that orthogonal weight among tasks can ensure the elimination of catastrophic forgetting. The article introduces a new orthogonal weight projection method for Task-Incremental Learning (TIL) to suppress catastrophic forgetting, referred to as Self-Adaptive Task Incremental Learning (STIL). First, STIL employs a similarity/dissimilarity task detector to assess the similarity between old and new tasks. When it detects that a new task is similar to old tasks, STIL can autonomously perform knowledge transfer. However, in cases where the new task is dissimilar to all old tasks, STIL switches to address catastrophic forgetting.

**Strengths:**

This article examines the limitations of the orthogonal gradient method and proposes a new approach involving orthogonal weight. Through theoretical analysis, the feasibility of orthogonal weight is confirmed. Additionally, the article introduces adaptive incremental learning, which better facilitates knowledge transfer and mitigates catastrophic forgetting.

**Weaknesses:**

Some of the formulas lack accompanying figures and textual explanations. Certain formula derivations and images from the supplementary materials are essential and should be incorporated into the main body of the text.

**Questions:**

1）In Eq.(9), two constraint conditions are imposed. Can convergence be achieved during the optimization process, or will it increase the optimization difficulty, potentially leading to optimization failure?
2）This paper introduces the method of orthogonal weights to mitigate catastrophic forgetting, but why does forgetting still occur, and what is the underlying reason?
3）Can the method proposed in this paper be applied to Class-Incremental scenarios?
4）Some crucial parts of the appendix should be incorporated into the main body of the text, such as Figure 2. Furthermore, it needs to be appropriately formatted to avoid potential misinterpretation.
5）The proposed Self-Adaptive Task Incremental Learning (STIL) should be placed before the introduction of weight orthogonality in the paper.

---

> ### Author Response · Authors · 2023-11-23
>
> Dear reviewer c582,
>
> Thanks for your valuable comments!
>
> **Q1**: Some of the formulas lack accompanying figures and textual explanations. Certain formula derivations and images from the supplementary materials are essential and should be incorporated into the main body of the text.
>
> **Response**: Thanks for your valuable comment! We have followed your suggestion and incorporated some crucial parts of the appendix, e.g., some figures and textual explanations, into the main body of our revised paper. Please see Figure 1 (a) - (b) and paragraph 1 of Sec. 4 in the revised paper.
>
> **Q2**: In Eq. (9), two constraint conditions are imposed. Can convergence be achieved during the optimization process, or will it increase the optimization difficulty, potentially leading to optimization failure?
>
> **Response**: As discussed in the second paragraph below Eq.(9) (corresponding to Eq.(8) of the revised paper), to avoid extensive computations in the constrained optimization and non-convergence in Eq.(8), considering the orthogonality of these two constraints and the contrastive loss, we convert Eq.(8) with the constraints as a new bi-objective loss Eq.(9), which eliminates the above issues that you are concerned about.
>
> **Q3**: This paper introduces the method of orthogonal weights to mitigate catastrophic forgetting, but why does forgetting still occur, and what is the underlying reason?
>
> **Response**: If the proposed method STIL only deals with forgetting without knowledge transfer (KT), then the orthogonal weights introduced in this paper can eliminate CF. However, as our STIL also carries out forward and backward KT operations, the backward KT of STIL would modify the weights of previous similar tasks resulting in slightly negative BWT on some datasets. Overcoming this issue is a further research direction.
>
> **Q4**: Can the method proposed in this paper be applied to Class-Incremental scenarios?
>
> **Response**: We believe that the proposed method STIL can also be applied to class-incremental learning (CIL) in those CIL methods that use a TIL method to fully overcome forgetting and a task label prediction to identify the correct task for each test sample.
>
> **Q5**: Some crucial parts of the appendix should be incorporated into the main body of the text, such as Figure 2. Furthermore, it needs to be appropriately formatted to avoid potential misinterpretation.
>
> **Response**: Thanks for your valuable comment! We have followed your suggestion and incorporated some crucial parts of the appendix into the main body in our revised paper. For example, we have put some important content that was originally in the appendix: Figure 1 (a)-(b) (Sec. 4), Theorem 5 (Sec. 3.2), ablation experiments of task similarity metric SDM (embedded in Table 3), etc., into the main body of the paper.
>
> **Q6**: The proposed Self-Adaptive Task Incremental Learning (STIL) should be placed before the introduction of weight orthogonality in the paper.
>
> **Response**: Since the proposed method STIL in this paper is based on the concept of Orthogonal Important Weight Subspace (OIWS), it is difficult to explain/discuss our STIL method without introducing the concept of OIWS first. That's why our paper content is organized this way.

---

### Official Review · Reviewer_bgzv · 2023-11-01

**Soundness:** 2 fair
**Presentation:** 1 poor
**Contribution:** 2 fair
**Rating:** 3
**Confidence:** 5

**Summary:**

This paper proposes an orthogonal (gradient) projection-based continual learning algorithm to address the over-consumption of network capacity per task and improve knowledge transfer.

In this method, the authors consider a task incremental learning setting to access the task boundaries and compute the weight similarity between tasks.

The proposed method extends gradient projection memory into a gradient update scheme, including the similar task which has positive transfer.

In the experiment, the proposed method STIL shows reasonably new scores in terms of average accuracy, but the backward transfer performance is somewhat limited.

**Strengths:**

The proposed method aims to address the following issues with orthogonal gradients:

- It cannot guarantee the elimination of catastrophic forgetting
- It limits knowledge transfer among gradient subspaces.

This paper combines gradient projection memory in terms of orthogonal important weight subspace and the log-barrier method of constrained optimization for knowledge transfer.

Through this approach, STIL can achieve better average test accuracy compared to baselines, including gradient projection memory.

**Weaknesses:**

The main concern I have on this paper is that the author’s claim has not been rigorously proved by theoretically and experimentally. The detailed comments on the above concern are as follows:

- Theorem 1 seems a trivial proposition, which states that the weight vector can be spanned by the graident vectors and the weght subspaces of each task are orthogonal. However, it cannot guarantee to explain the property of catastrophic forgetting in the final logit layer of DNN, which is directly connected to the output prediciton.
- The proposed $\mathcal{L}_{sim}$ is based on the constrained optimization, which cannot be applied to the nonconvex domain (most deep learning probelms) directly. In addition, the orthogonality of weight matrix cannot explain the orthogonality of the hidden layers with non-linear activation, so I think the author should provide more empirical evidence why this proposed loss facilitate knowldedge transfer.
- The similarity detection based algorithm seems not novel enough to explain the task distribution distance by measuring on the online data stream. The threshold $\theta$ is too heuristic without any empirical explanation.

I think that this paper is a simple combination of the existing gradient projection memory method and the heuristic similiarity based update rule.

Updating scheme among orthogonal weight subspaces can be a interesting work, but this paper does not theoretically or empirically provides what happens in the non-linear activation layer.

Furthermore, the proposed method should have acheived the best performance in the both metrics, test accuracy and BWT simultaneously if the propsed pipeline succesfully works to handle catastrophic forgetting and knowledge transfer simultaneously.

## Miscelleaous

The main idea section is not well written and easy to follow.

First of all, some main statement is not clear as follows:

- In Equation (1), the subscription of the weight matrix is the task, so it does not fit the SGD update definition which is on the time-step.
- The main terminology, OWIS is not explained in the paper including supplementary materials.
- The pipeline figure and the pseudocode are in supplementary material, so it is hard to understand how the proposed algorithm works.
- There are two many aabrevation and terminology without enough explanation such as knowledge base in page 5.
- I think this method is the extension fo GPM, but the discussion on the comparision in the experimental section is not enough. For example, why STIL without OIWS  slightly performs better than GPM.

**Questions:**

I think that the propsed method shows a reasonable performance in several benchmarks, but it is more important to address the funtamental principle of continual learning with deep neural networks.

As the paper, gradient projection memory, provides the figure of interference activations with several threshold, can the author provide an **empirical materials** what the proposed algorithm does something to prevent catastrophic forgettting and increase knowledge transfer.

The second question is that what is the main difference between gradient projection memory and the OIWS-based til strategy.

---

> ### Author Response · Authors · 2023-11-23
>
> Dear reviewer bgzv,
>
> Thanks for your valuable comments!
>
> **Q1**: Theorem 1 cannot guarantee to explain the property of catastrophic forgetting (CF) in the final logit layer of DNN, which is directly connected to the output prediction.
>
> **Response**: The proposed method STIL is the method of updating/learning the neural network weights from the first layer of the network to the last logical layer with the following Theorem 1 (see Eq.(7). The property of overcoming CF in the last logit layer of DNN in Theorem 1 has been verified by the experimental results in Tables 1-2.
>
> **Q2**: The proposed $\mathcal{L}_{sim}$ is based on the constrained optimization, which cannot be applied to the nonconvex domain (most deep learning problems) directly...
>
> **Response**: So far, there is no evidence that the proposed $\mathcal{L}_{sim}$ cannot be applied to the nonconvex domain (...) directly. The neural network architecture with nonlinear activation layer (ReLU) used in our experiments, such as AlexNet and ResNet-18, all work well with our proposed method STIL. Unlike the existing methods, this paper provides a total of 11 various CL task sequences to fully evaluate the effectiveness of our method. Please see the experiment results in Tables 1-2 and Table 11 in Appendix D.4.
>
> **Q3**: The similarity detection-based algorithm seems not novel ......
>
> **Response**: We proposed the task similarity detection method based on the task distribution Wasserstein distance for the first time, and the experimental results (see Table 3) show that this method is effective and accurate. For the theoretical basis and empirical explanation of the value of threshold $\theta$, refer to its explanation in Eq. (11). For its experimental values in different datasets, refer to Table 4 in Appendix B.
>
> **Q4**: I think that this paper is a simple combination of the existing gradient projection memory method and the heuristic similarity-based update rule.
>
> **Response**: There is misunderstanding. We aim to to overcome the three main weaknesses of the existing OG-based methods. This paper gives a novel and effective solution, please refer to Section 1 Introduction. Specifically, we first theoretically showed that the existing OG-based methods can only mitigates CF but cannot eliminate CF (Theorem 1). Then, we introduce the new concept OIWS and propose an OIWS-based TIL strategy, which can solve the problems (1) and (2) (see Theorems 1-3, Lemma 1, Table 1 and Figure 6 in Appendix). Based our theoretical research for KT, we give the bounds that can lead to negative FKT and BKT. Moreover, we propose a new task similarity detection method based on only direct reuse of the knowledge already learned in the current network and a new cross-entropy loss with constraints along with its optimal solution, which efficiently and effectively solve the problem (3) (see Table 2).
>
> **Q5**: For updating scheme among orthogonal weight subspaces, this paper does not theoretically or empirically provide what happens in the non-linear activation layer.
>
> **Response**: The proposed OIWS-based method works only on each parameter or weight layer of the neural network (see Eq. (7)), and the gradient calculation and updating of the nonlinear activation is taken care of by optimizer SGD of the deep machine learning platform.
>
> **Q6**: The proposed method should have achieved the best performance in the both metrics, test accuracy and BWT simultaneously......
>
> **Response**: The ACC performance of a TIL method is determined by its abilities to overcome forgetting, FKT, and BKT. The proposed STIL method is superior to existing strong baselines in resisting forgetting, FKT, and BKT. Please refer to the experimental results shown in Tables 1-2.
>
> **Q7**: I think this method is the extension of GPM, but the discussion on the comparison in the experimental section is not enough......
>
> **Response**: In our revised paper, we have highlighted the main difference between our STIL and the method GPM in the resubmitted/revised paper (see paragraph 2 in Sec. 5.2). Briefly, as GPM has no explicit KT mechanism or OIWS-based strategy for CF prevention,  our STIL achieves a better performance than GPM.
>
> **Q8**: Can the author provide an empirical material what the proposed algorithm does something to prevent CF and increase KT.
>
> **Response**: We don't quite understand your ''empirical material''. We have conducted extensive empirical experiments and reported the results in Tables 1-2 and Table 11 in Appendix D.4 to demonstrate the effectiveness of our STIL in resisting forgetting and transferring knowledge.
>
> **Q9**: What is the main difference between GPM and the OIWS-based strategy?
>
> **Response**: The main differences are: 1) STIL satisfies the two necessary and sufficient conditions of resistance to forgetting stated in Theorem 1, while the GPM method only satisfies condition 1 without guarantee condition 2; 2) STIL has a knowledge transfer (KT) mechanism, while GPM has no explicit KT mechanism.

---

> ### Comment · Reviewer_bgzv · 2023-11-28
>
> **Q1, Q5, Q7, Q9**
>
> Thanks for the response. I have already been aware that the proposed method is more useful than the baselines empirically, but the orthogonal subspace based continual learning methods [Orthog-subspace (Chaudhry et. al. 2020), GPM] have already studied with the simple and insightful theoretical backgrounds.
>
> As far as I understand, Theorem 1 can explain the case of a single layer networks without nonlinear activation because the propagated (multiplicated) hidden layer values for multi-layer NNs with nonlinear activation can loss the orthogonlity property by non-linear activation function.
>
> To propose more meaningful theoretical statement for Theorem 1 for this venue, I think that the author should consider the above scenario. This is why I mentioned the output prediction logit.
>
> **Q2, Q8**
>
> For Q2, I partially agree with the author’s respose that the current researches cannot explain the convergence property of noncovex continual learning perfectly. However, in Section 4.2, the proposed method use a log-barrier method which is not directly applicable for nonconvex case without any theoretical justification (this is for convex problem).
>
> For the nonconvex case, the regularization method is a well-known method to relax the constrained problem for nonconvex case, but the log-barrier is not.
>
> I think the author should why the log-barrier method work better with insightful figures with empirical results at least.
>
> **Q3, Q4, Q6**
>
> Thank you for the feedback.
>
> I have already seen that the propsed method shown reasonable perfomance among various CL benchmarks.
>
> However, the proposed method seems a heuristic combination of existing methods, so the author should explain why this combination have a notable effect on continual learning with figures.
>
> For example, the GPM paper describes their claim by demonstrating figure 3 which explains interference activations.

---

### Meta-Review · Area_Chair_NENK · 2023-11-30

**Metareview:**

**Summary**: This paper studies catastrophic forgetting in continual learning, showing theoretically that prior methods based on orthogonal gradients are insufficient. The paper then proposes a method using weights that are orthogonal for different tasks.

**Strengths**: Reviewers appreciated the theoretical contribution of the paper together with it's "comprehensive experiments." They remarked on the good ablation experiments as well.

**Weaknesses**: The main concerns about the paper center on the theoretical results, and whether they support the claims made in the paper and observed in the empirical experiments. There were also some concerns about what a ``gradient'' space is.

**Justification For Why Not Higher Score:**

While the empirical results seem strong, the some issues with the theory make me wary of accepting this paper.

**Justification For Why Not Lower Score:**

N/A

---

### Decision · Program_Chairs · 2024-01-16

Reject